# NTPDase1/CD39 Ectonucleotidase Is Necessary for Normal Arterial Diameter Adaptation to Flow

**DOI:** 10.3390/ijms242015038

**Published:** 2023-10-10

**Authors:** Julie Favre, Charlotte Roy, Anne-Laure Guihot, Annick Drouin, Manon Laprise, Marc-Antoine Gillis, Simon C. Robson, Eric Thorin, Jean Sévigny, Daniel Henrion, Gilles Kauffenstein

**Affiliations:** 1MITOVASC Institute, CARFI Facility, CNRS UMR 6015, INSERM U1083, Angers University, 49045 Angers, France; j.favre@unistra.fr (J.F.); daniel.henrion@univ-angers.fr (D.H.); 2Montreal Heart Institute, Department of Surgery, Université de Montréal, Montreal, QC H1T 1C8, Canada; 3Animal Physiology Service, Institut de Recherches Cliniques de Montreal (IRCM), Montreal, QC H2W 1R7, Canada; manon.laprise@ircm.qc.ca; 4Department of Medicine, Division of Gastroenterology, Beth Israel Deaconess Medical Center, Harvard Medical School, Boston, MA 02215, USA; 5Centre de Recherche du CHU de Québec, Université Laval, Québec City, QC G1V 4G2, Canada; 6Département de Microbiologie-Infectiologie et D’immunologie, Faculté de Médecine, Université Laval, Quebec City, QC G1V 0A6, Canada; 7INSERM UMR 1260—Regenerative Nanomedicine, CRBS, Strasbourg University, 67000 Strasbourg, France

**Keywords:** NTPDase1/CD39, ATP, flow-mediated vasodilation, shear stress, vascular remodeling

## Abstract

NTPDase1/CD39, the major vascular ectonucleotidase, exerts thrombo-immunoregulatory function by controlling endothelial P2 receptor activation. Despite the well-described release of ATP from endothelial cells, few data are available regarding the potential role of CD39 as a regulator of arterial diameter. We thus investigated the contribution of CD39 in short-term diameter adaptation and long-term arterial remodeling in response to flow using *Entpd1^−/−^* male mice. Compared to wild-type littermates, endothelial-dependent relaxation was modified in *Entpd1^−/−^* mice. Specifically, the vasorelaxation in response to ATP was potentiated in both conductance (aorta) and small resistance (mesenteric and coronary) arteries. By contrast, the relaxing responses to acetylcholine were supra-normalized in thoracic aortas while decreased in resistance arteries from *Entpd1^−/−^* mice. Acute flow-mediated dilation, measured via pressure myography, was dramatically diminished and outward remodeling induced by in vivo chronic increased shear stress was altered in the mesenteric resistance arteries isolated from *Entpd1^−/−^* mice compared to wild-types. Finally, changes in vascular reactivity in *Entpd1^−/−^* mice were also evidenced by a decrease in the coronary output measured in isolated perfused hearts compared to the wild-type mice. Our results highlight a key regulatory role for purinergic signaling and CD39 in endothelium-dependent short- and long-term arterial diameter adaptation to increased flow.

## 1. Introduction

The fine regulation of blood flow via neurogenic, hormonal and mechanical means allows for a proper organ perfusion and an adapted response to metabolic demand. Most, if not all, pathologies presenting with a vascular component are characterized by an endothelial dysfunction, which originates from deleterious hypertrophic and fibrotic vascular remodeling, contributing to pathologies such as hypertension, atherosclerosis, and heart failure. It is well accepted that flow-mediated vasodilation (FMD) reflects endothelium-dependent and mostly nitric oxide (NO)-mediated arterial function and constitutes a surrogate marker of vascular health that exceeds the predictive value of traditional risk factors [1,2,3]. The mechanisms involved in cellular shear stress sensing and FMD are complex. They involve specific structures such as the endothelial cilium, glycocalyx, and caveolae, as well as the molecular entities implicated in mechanical forces’ perception, signal transduction, and adapted cellular response [4]. Due to the link between endothelial dysfunction and cardiovascular pathologies, knowledge of the mechanisms underlying the endothelial vasculoprotective response to flow represents a major therapeutic opportunity [5]. Researches on the cellular and molecular mechanisms involved in FMD have been the subject of (sometimes divergent) advances in recent years, particularly in regard to the involvement of purinergic signaling (i.e., signaling via extracellular nucleotides). 

Purinergic signaling is a key component of vascular homeostasis, which is well documented as contributing to platelet aggregation, inflammation, and endothelial permeability [6]. Moreover, extracellular nucleotides participate in the local control of blood flow through binding to smooth muscle and endothelial purinergic P2 receptors. The activation of vascular smooth muscle cell (VSMCs) P2 receptors promotes constriction via P2X or pyrimidine-sensitive (UTP/UDP) P2Y receptors that take part in the neurogenic response of resistance arteries. By contrast, the activation of endothelial P2 receptors induces vasorelaxation via the production of nitric oxide (NO), prostacyclin, and the endothelium-derived hyperpolarizing factor [7].

Many data support ATP as a mediator of the vasodilatory response to shear stress, which constitutes a major stimulus to the endothelium [8]. Vascular endothelial cells (EC) are long known to release ATP in response to mechanical forces; in particular, the shear forces exerted by the circulating blood at their surface [9]. Several studies suggest that this ATP may participate in the mechanism of FMD via an autocrine activation of the endothelial P2 receptors. The importance of this mechanism is underlined by the altered adaptation of the arterial diameter in response to the acute and chronic change in flow when genetically or pharmacologically interfering with the ion gated P2X4 (ATP) receptor [10] or G-protein coupled receptors P2Y_2_ (ATP/UTP) [11,12] and P2Y_1_ (ADP) receptors [13,14]. Hence, considering its role as an actor of vascular tone adaptation, the necessity to keep operative endothelial signaling for ATP and other nucleotides is of particular importance for arterial diameter adaptation in pathophysiological conditions. 

The nucleoside triphosphate diphosphohydrolase-1 (NTPase1, aka CD39) is the major vascular ectonucleotidase that regulates the extracellular di and triphosphonucleoside concentration, thus controlling P2 nucleotide receptor activation at the surface of the vascular endothelium and the blood circulating cells. Invalidation of the *Entpd1* gene in mice unveiled its role in preventing platelet-dependent thrombosis [15,16], vascular permeability, leukocyte extravasation, and tissue damage after ischemia via the combination of nucleotide hydrolysis and adenosine generation [17,18,19]. Importantly, the decrease in CD39 expression in the atheroprone arterial area was associated with endothelial dysfunction and accelerated immune cell infiltration, contributing to atherosclerotic plaque development and suggesting a key contribution of the enzyme in the preservation of endothelial homeostasis [20].

The growing body of evidence points to extracellular ATP as a key mediator in the control of endothelial homeostasis in general and shear stress mechanosensing in particular. However, no data regarding the role of ectonucleotidases in flow-mediated arterial remodeling have been reported. We previously showed that CD39 limits the nucleotide-dependent vasodilation [21]. In the present work, we investigated the impact of CD39 deletion on endothelial responses in conduit and resistance arteries in which we focused on FMD and arterial adaptation to chronic change in shear stress. 

## 2. Results

### 2.1. Impact of CD39 Deletion on Hemodynamic Parameters

We assessed whether CD39 deletion had an impact on the global hemodynamic parameters in conscious (Table 1) and anesthetized mice (Table 2). Significant differences could be evidenced via radiotelemetry in conscious animals: decreased systolic blood pressure (SBP) and pulse pressure (PP) and lower heart rate in *Entpd1^−/−^* mice compared to WT (Table 1 and Appendix A). Anesthetized mutant mice did not a show significant SBP decrease while PP was significantly reduced. By contrast, the heart rate was increased in *Entpd1^−/−^* mice (Table 2). Hence, the PP decrease observed both in awake and anesthetized *Entpd1^−/−^* animals, constitutes the most consistent observation. The PP is influenced by arterial stiffness among other parameters including blood pressure. Our results thus suggest that mutant arteries may display increased arterial compliance. Of note, we could not evidence any differences in the mechanical properties of isolated pressurized MRA ex vivo nor any histomorphological changes (Appendix A) and fibrosis in larger vessels (Appendix A). Considering the differences between conscious and anesthetized animals, in particular, the measurement of the heart rate, we may hypothesize that these may reveal a contribution of the enzyme in nerve action potential propagation and cardiac electrical conduction. Indeed, the neuronal expression of CD39 has been reported in porcine heart [22], as well as its contribution in norepinephrine release from cardiac sympathetic nerves, revealing a contribution of CD39 and purinergic signaling in adrenergic activity [23]. 

### 2.2. Impact of CD39 Deletion on Vasoreactivity of Conductance and Resistance Arteries

There was no significant difference between groups in the contractile response to Phe in aortic and mesenteric arteries, as previously observed [24] (Figure 1). By contrast, a significant increase in sensitivity to serotonin was observed in coronary arteries isolated from *Entpd1^−/−^* mice (Table 3) without changes in the maximal contractile response compared to *Entpd1^+/+^* mice (Figure 1C and Table 3). 

Cumulative concentrations of ATP induced a marked relaxing response in all three types of vessels (Figure 2A–C) isolated from *Entpd1^−/−^* mice with a bimodal response in aorta and MRA, which contracted at higher ATP concentrations. Conversely, only a slight vascular response was observed in wild-type littermates, ranking from a limited relaxing response in LCA to a contraction at high concentrations of ATP in MRA and a heterogeneous response in ThAo. 

In view of these altered responses to ATP, we next evaluated the endothelium-dependent relaxation in response to Ach. We found that CD39 deletion differentially affected the vascular response to Ach in large and small arteries (Figure 3). 

Figure 3A represents the endothelial-dependent relaxing response to Ach, a muscarinic agonist. A slight but significant increase in relaxation to Ach was measured in *Entpd1^−/−^* aortas (10^−6^ M, Figure 3A) despite no significant difference in EC_50_ compared to wild-type arteries (Table 3). The blockade of NO production by LNAME similarly abolished the relaxations in both groups (Figure 3B) and no changes in the endothelium-independent response to the NO donor SNP were observed (Figure 3C), suggesting a higher NO response to Ach in *Entpd1^−/−^* aortas as previously observed elsewhere [25]. 

Conversely, endothelium-dependent relaxation to Ach was significantly altered in MRA from *Entpd1^−/−^* mice compared to the controls (Figure 3A, middle), with no change in the maximal relaxation (Table 3). This alteration was still observed in the presence of the NO synthase inhibitor (LNAME), but normalized via an additional cyclooxygenase inhibition (INDO). No differences in the response to the NO donor SNP were observed (Figure 3B, middle). This suggests an apparent endothelial dysfunction in *Entpd1^−/−^* mesenteric arteries which involves a COX-related endothelial-derived relaxing factor, likely pointing to PGI_2_/prostacyclin. 

In coronary arteries, CD39 deficiency slightly affected the relaxation induced by Ach, while LNAME abolished the relaxations in both *Entpd1^−/−^* and WT mice (Figure 3B, right), suggesting an alteration in the NO-dependent relaxation. Endothelium-independent relaxations to SNP were equivalent between groups in all types of arteries studied (Figure 3C).

### 2.3. Impact of Entpd1 Deletion on Microvascular Response to Mechanical Stimuli

We assessed the arterial dilatation of MRA by increasing the intraluminal flow either acutely ex vivo or chronically in vivo. While the MRA isolated from *Entpd1^+/+^* mice dilated in response to the acute stepwise increase in the intraluminal flow, MRA isolated from *Entpd1^−/−^* mice barely showed any increase in the internal diameter (% maximal dilation at flow 150 µL/min: *Entpd1^−/−^* 7.82 ± 0.67 vs. *Entpd1^+/+^* 47.97 ± 2.85; *p* < 0.001) (Figure 4A). 

As several nucleotide receptors were proposed to contribute to FMD, we evaluated the expression of the dominant vascular P2 receptors in MRA via RTqPCR (Appendix A). The results evidenced equivalent mRNA levels in both genotypes except for the P2Y_2_ receptor, which appeared to be 2-fold increased in the mutant mice (0.071 ± 0.006 vs. 0.126 ± 0.011 for *Entpd1^+/+^* and *Entpd1^−/−^*, n = 5 and 3 respectively; *p* = 0.0238 Mann–Whitney test).

The in vivo ligation of two adjacent mesenteric arteries (Low flow, LF) led to an artificial chronic increase in the blood flow in the central artery (High flow, HF) (scheme in Figure 4B). As expected in this model [26], after 2 weeks, we measured a higher diameter in the HF mesenteric arteries isolated from WT mice compared to the control arteries (Normal Flow, NF Figure 4C). However, this outward remodeling was absent in the arteries from *Entpd1^−/−^* mice (Figure 4D). Of note, equivalent inward remodeling in the LF arteries occurred both in WT and *Entpd1^−/−^* mice.

In the isolated perfused heart, we evidenced an alteration in the coronary flow in *Entpd1^−/−^* compared to the wild-type mice (Figure 5A). This defect was also observed after artificially increasing the heart rate via cardiac pacing (Figure 5A,B). Apart from the coronary output, no major modification of cardiac function could be detected in the isolated hearts of *Entpd1^−/−^* mice, despite a slight increase in Pmax and dP/dt max, suggesting a better left ventricular contraction both at rest and in the paced conditions (Figure 5C).

## 3. Discussion

Structural adaptation of the vasculature in response to the increased flow condition is required to fulfill and adapt to the physiologic and metabolic demand [27] in contexts as varied as growth, exercise, pregnancy, and cancer. Several recent works highlighted the importance of ATP and the contribution of specific P2 receptors in the arterial diameter adaptation in response to the acute and chronic change in the blood flow; however, very few studies focused on the role of extracellular nucleotide hydrolysis in these processes. Here, we evidenced that CD39, the dominant vascular ectonucleotidase, controls the ATP-dependent relaxation, and its absence results in an impaired ability to adapt the arterial diameter to the acute and chronic increase in flow.

We found that ex vivo vasorelaxation in response to ATP was greatly enhanced in both the resistance and conductance arteries of *Entpd1^−/−^* mice, pointing to the extreme efficacy with which the enzyme modulates endothelial purinergic signaling. Additional experiments evidenced slight modifications in the pharmacological responses to non-nucleotide agonists such as Ach and serotonin, likely pointing to a global alteration and/or adaptation to vasoactive mediators’ responsiveness. Most importantly, a marked dysfunction was evidenced when evaluating the adaptation of the resistant artery diameter in response to the increased flow, pointing to a role of the enzyme in shear stress mechanotransduction. 

When investigating the reactivity of *Entpd1^−/−^* mouse arteries, a better sensitivity to serotonin could be assessed in the coronary arteries and thoracic aorta (Appendix A). Such enhanced contraction in the *Entpd1^−/−^* arteries may be due to the release of nucleotides following 5HT stimulation, leading to autocrine amplification. Indeed, a secondary pannexin1-dependent ATP release following the serotoninergic/adrenergic stimuli was already documented [28,29], and a potential crosstalk between the serotoninergic and purinergic systems has been suggested in *Entpd1^−/−^* macrophages [30].

We previously showed that greater P2 receptor reactivity exacerbated relaxing responses in *Entpd1^−/−^* animals [21]. We extended these observations to small diameter MRA and LCA. While the cumulative concentrations of ATP induced a moderate contractile response in the aorta and MRA of WT mice, CD39 deletion unveiled a marked relaxing response to ATP (Figure 1), confirming that the enzyme limits endothelial ATP receptor activation ex vivo [21,31]. Our observations also revealed a supra-relaxing response to submaximal Ach concentrations in the aorta of *Entpd1^−/−^* animals, corroborating with previous findings [25]. Although the over-expression of eNOS was pointed out by the authors, we could not evidence the significant increase in eNOS expression via western blot (Appendix A) nor via RTqPCR (Appendix A) in the *Entpd1^−/−^* aorta. Other mouse models presenting ATP signaling deficiency with global or endothelial P2Y_2_ deficiency did not show any differences in the aortic response to Ach [11,31]. Still, a decrease in eNOS expression was measured in the aortic arch of endothelial *P2ry2^−/−^* mice [32]. Altogether, these data point to a possible feedback loop controlling the eNOS expression/function in animals with an alteration in endothelial ATP signaling.

Conversely, MRA and LCA from *Entpd1^−/−^* mice showed a slight alteration in the submaximal responses to Ach, suggesting an endothelial dysfunction in the small diameter resistance arteries. Indeed, the response to the NO donor SNP was unchanged, evidencing a normal smooth muscle cell response. In MRA, this alteration seems to rely on the impaired PGI_2_ relaxing component. Of note, ATP has already been reported to influence the PGI_2_ release [33]. By contrast, NO-dependent relaxation was impaired in LCA. In these arteries, the endothelial response is mostly dependent on the NO production [34]. Although these changes are modest, they suggest that CD39 impacts endothelium-dependent vasodilation in the whole arterial system with a differential effect, depending on the contribution of the endothelium-derived relaxing factor in these vessels. 

In contrast with the slight shift in response to Ach, FMD was virtually abolished in MRA of *Entpd1^−/−^* mice (Figure 4). It is long known that the mechanical constraints, endothelial shear stress in particular, drive the cellular nucleotide release [9]. An enlarging body of evidence points to the importance of ATP and the contribution of specific P2 receptors in FMD. This hypothesis is supported by the fact that the adaptation of the arterial diameter in response to the acute and chronic changes in flow is altered when genetically or pharmacologically interfering with P2X4 (ATP) [10], P2Y_2_ (ATP/UTP) [11,12], or P2Y_1_ (ADP) receptors [13,14]. Our study confirms, through the prism of their hydrolysis, that nucleotides are key actors of endothelial mechanotransduction. Recently, Stephen Offermanns’ group proposed that shear stress-mediated endothelial ATP release depends on the mechanosensitive channel Piezo-1 and induces the activation of the P2Y_2_ receptor/G_q/11_ protein pathway [11,12]. This is in good agreement with the data reporting the involvement of the G_q_ protein in flow mechanosensing processes [35]. Invalidation of one of these three partners (Piezo-1, G_q,11_, P2Y_2_), specifically in the endothelium, leads to a systemic, but reversible, increase in blood pressure [11,12]. The model proposed by Wang et al. diverges from that proposed by Ando’s group in which ATP is released by plasma membrane mitochondrial F1/F0 ATPase, establishing a link between FMD and the production and release of ATP by endothelial mitochondria [36]. Such a mechanism has been suggested in a model highlighting the shear stress-induced autophagy as a starter point to provide fuel (glucose) to the mitochondria, which in turn increases the excretion of ATP and ADP by ecto-F1-ATPase to finally activate endothelial P2Y_1_ and promote eNOS activation [13]. The latter study was performed on bovine endothelial cells after 3 h of shear stress, but in a more recent study, the authors showed a defect in the acute FMD of femoral arteries isolated from *P2ry1^−/−^* mice [14]. The relative contribution of the (P2Y_2_, P2Y_1_, P2X4, and P2X1) receptors and/or their interaction in FMD remains to be determined. 

Interestingly, we evidenced a 2-fold increase in the P2Y_2_ receptor in *Entpd1^−/−^* mice MRA. This observation suggests a potential relationship between the ATP bioavailability and P2Y_2_ receptor expression. In addition, considering that P2Y_2_ is a key actor of FMD, its over-expression in *Entpd1^−/−^* MRA can appear as a compensatory mechanism that one can have expected to rescue the FMD response. Hence, the fact that FMD is nevertheless abolished in *Entpd1^−/−^* vessels strengthens the importance of the CD39-dependent regulation of the P2Y_2_ ligand’s abundance, rather than the expression of the receptor itself.

Importantly, it may appear contradictory that, at the same time, CD39 deletion unmasks potent responses to exogenous ATP and impairs FMD that relies on the endogenous secretion of ATP. One way of explaining this apparent paradox is to consider the protective role of CD39 against P2 receptor desensitization. Indeed, several P2 receptors present a refractory state following the exposure to their agonist. The mechanisms involved have been well documented for P2X1 and P2Y_1_ receptors, which present a high propensity to desensitize following the ATP and ADP exposure, respectively. Even the relative contribution of the endothelial P2 receptors in FMD remains to be determined, as our results suggest the contribution of “desensitizing” P2 receptor(s). An alternative explanation (independent of the desensitizing process) is to consider that CD39 is required to “reset” P2 receptor activability, by preventing their chronic activation and preserving full ATP (or ADP) responses. Corroborating this hypothesis, in isolated rat mesenteric arteries, ATP, such as its non-hydrolysable analogue ATPγS, were shown to potentiate FMD, whereas in the presence of the ectonucleotidase inhibitor, ATP was less effective with repetitive flow stimulations [37]. Also, it is not excluded that the contribution of different receptors occurs with specific kinetics. Overall, our data corroborate the above-mentioned works and show that CD39 helps in regulating FMD and arteriolar remodeling by preserving the endothelial P2 receptor functionality. In line with the need for an adequate NTPDase activity to maintain the P2 receptor sensitivity, we evidenced in a previous work that a specific level of apyrase (CD39 mimetic) was required to maintain the full functionality of P2X1 receptors in vas deferens smooth muscle cells [38].

Interestingly, in HUVEC submitted to shear stress, an active and sustained release of ectonucleotidases, with CD39 and CD73-like activities, is associated with a dramatic increase in ATPase activity and a rise in adenosine formation in the extracellular medium [39]. This enzymatic activity participates in the rapid decrease in ATP (within minutes) after shear stress initiation and certainly helps in preventing receptor desensitization. In a recent study, the circulating levels of CD39 were linked to decreased myocardial blood flow in humans [40]. These data imply that the pathological shedding/loss of endothelial ectonucleotidases is linked to altered myocardial blood flow.

Importantly, *Entpd1^−/−^* mice also exhibited a defect in the long term vascular adaptation to a chronic increase in blood flow. We documented the link existing between the acute vascular response to flow and long-term outward remodeling [41]. Our present data thus suggest that CD39 and local purinergic signaling are required to maintain an acute response to flow and ensure that the sequential events allow for long term arterial wall remodeling. 

Arterial remodeling involves an inflammatory process, the perivascular infiltration of lymphocytes and macrophages, and extracellular matrix reorganization [42,43]. The mechanisms involved in the dilation and the local inflammatory process are not necessarily superimposed. Indeed, the defect of adaptive remodeling in *Entpd1^−/−^* mice can result from the alteration in the acute response of the endothelial cells to flow, but also from an impairment of the recruitment of inflammatory cells. However, leukocyte infiltration is rather increased in *Entpd1^−/−^* tissues in ischemic conditions [18,44]. Additionally, we did not measure significant differences in periaortic fat leukocytes (CD45^+^) between wild-type and *Entpd1^−/−^* mice via immunofluorescence (Appendix A) or RTqPCR (Appendix A). The expression of genes involved in the oxidative stress and matrix remodeling was also not altered in the *Entpd1^−/−^* aorta. Altogether, these data do not argue in favor of a deficient immune infiltration causing altered outward remodeling in CD39 deficient animals, but rather point to a defective endothelial response to shear stress. 

These observations may somehow be corroborated with those obtained with *P2rx4^−/−^* mice, which develop an impaired inward arterial remodeling after carotid ligation [10]. Even the mechanisms involved in outward eutrophic and those underlying inward hypertrophic remodeling do not fully overlap; they share common traits and our data suggest that altered P2X4 receptor functionality could account for the impaired arterial remodeling in *Entpd1^−/−^* arteries. To our knowledge, long term vascular remodeling has not been investigated in *P2ry2^−/−^* animals. Noteworthy, we were unable to demonstrate any differences in the morphology or viscoelastic properties of MRA between wild-type and *Entpd1^−/−^*, excluding the structural alterations beneath the defective arterial adaptation to blood flow (Appendix A). 

The evaluation of the basal coronary function in a more integrated setup in the isolated perfused hearts revealed a reduced coronary output in *Entpd1^−/−^* mice. This defect was observed even when the stimulated flow was increased via the elevated cardiac frequency on paced isolated hearts (Figure 5). Cardiac functional parameters revealed that isolated *Entpd1^−/−^* hearts developed a better contraction than the WT hearts. Although we cannot exclude an impact of the ventricular contractile function, the decrease in coronary flow reflects an increased coronary vascular tone, which probably results from a loss of the flow-mediated coronary dilatation and a possible higher vasoconstrictive state. Indeed, we previously reported an exacerbated contractile response associated with the preservation of a higher sensitivity to the P2Y_6_ agonist in *Entpd1^−/−^* mice, with direct repercussions showing as an exacerbated myogenic and possibly a neurogenic constriction [24,45]. Although these investigations were performed ex vivo, one can hypothesized that such an exacerbated tone may impact the coronary blood flow in vivo through increasing microvascular resistance.

Interestingly, the P2Y_2_ and P2Y_6_ receptors were shown to play a distinct role in the coronary vascular tree. While endothelial P2Y_2_ would mediate the UTP-mediated vasodilation of large proximal arteries, the going-with-the-flow hydrolysis of this uracil nucleotide into UDP would be responsible for the vasodilation of the downstream microvessels, as P2Y_6_ seems to be preferentially expressed in the distal coronary endothelium [46]. Thus, in the model of Haanes and colleagues, endothelial CD39 plays a pivotal role in providing the adequate vasodilating agent to the appropriate sites. 

The decrease in the coronary output in *Entpd1^−/−^* mice could also be attributed to a decrease in the coronary reserve, although, in this case, there is no histological data supporting this hypothesis, with the morphology of large coronary arteries being reported equivalent to the wild-types [47]. Nevertheless, considering the importance of adenosine signaling in the heart [48], we may also hypothesize that CD39 deficiency decreases the formation of AMP, and thus, the local amount of adenosine. Interestingly, perfused hearts isolated from *cd73^−/−^* mice, which are deficient for ecto-5′-nucleotidase, show a basal decrease in the coronary flow while the in vivo hemodynamic and cardiac parameters remained normal [49]. Noteworthy, the studies on isolated rat hearts demonstrated that the cardiac hydrolysis of intravascular ATP is very efficient, which certainly explains the lack of desensitization in response to ATP, ADP, and ADO [50].

Apart from the direct effect of flow on the endothelial ATP release, shear stress has been shown to modulate CD39 expression. Indeed, while CD39 was down-regulated at the site of the disturbed flow in the mouse aortic arch, in vitro studies evidenced that its expression was increased in response to the laminar shear stress on endothelial cells [20]. Moreover, we have reported a down-regulation of CD39 activity in experimental hypertension [51]. Considering that ATP levels are increased in hypertension [52], the CD39 decrease could be a major determinant of the alterations of the flow-mediated response and remodeling in the microvascular territory associated with this pathology. 

The use of potato apyrase (which mimics CD39 activity) has already shown the potential impact of NTPDase1 on the endothelial response to flow. Indeed, different studies depicted, with different kinetics on the cultured endothelial cells, the lack of a calcium response or eNOS activation to shear stress by artificially increasing CD39 activity [11,13,53]. Considering these observations, we could have expected a potentiated endothelial response to shear stress in *Entpd1^−/−^* mice. However, as previously observed, with regard to the platelet function of *Entpd1^−/−^* mice [15], the global deletion of CD39 leads to a paradoxical phenotype resembling a reinforcement of the enzyme function. Such counter-intuitive observations were reported in a permanent coronary ligation model of myocardial infarction where *Entpd1^−/−^* mice showed less mortality and cardiac rupture [47]. In the same way, *Entpd1^−/−^/ApoE^−/−^* mice developed less atherosclerotic lesions [54]. However, without abolishing CD39 expression, its down-regulation in heterozygote *Entpd1^+/−^* mice led to the worsening of atherosclerotic plaques or platelet aggregability [15,20] and then unveiled a protective role of the enzyme in these pathologies. 

To our knowledge, no data are available concerning the role of platelets on acute FMD or chronic outward arterial remodeling. Moreover, most of our data were obtained in vitro, which excludes the contribution of platelets in the differences we observed in *Entpd1^−/−^* mice.

Anormal adaptative vascular remodeling, including proliferative hypertrophy and deficient angiogenesis, has already been reported in *Entpd1^−/−^* mice. Notably, a hyperproliferative phenotype could be evidenced in the pathological models of pulmonary hypertension [55] and carotid restenosis [56,57]. It has been hypothesized that this response may be due to the unleashed VSMC P2 receptor-dependent proliferation. Taking into consideration that smooth muscle cell proliferation is tonically repressed by endothelium-derived factors, in particular NO [58], we hypothesize that these exacerbated hypertrophic/proliferative responses previously reported in *Entpd1^−/−^* animals may be due to impaired endothelial flow sensing. 

Impaired angiogenesis has also been reported in hypoxic and tumoral angiogenesis in *Entpd1^−/−^* mice [30,59]. The use of knock-out mice revealed that, via the regulation of local purinergic signaling, CD39 modulates many different cardiovascular pathology manifestations. Of vascular importance, CD39 global deletion has been shown to lead to diabetes or to worsen cardiac ischemia-reperfusion outcomes [60]. The selective targeting of CD39 deletion in the endothelium could certainly help in deciphering the specific role of endothelial CD39 versus its immune face in an in vivo pathological context. The extent in which the deficit in endothelial shear sensing we report here takes part and contributes to these pathological traits and metabolic affect may deserve specific attention. 

## 4. Materials and Methods

### 4.1. Animals

All animals were manipulated in accordance with the European Community Standards on the Care and Use of Laboratory Animals (Ministère de l’Agriculture, France, authorization No. 6422). Experimental protocols were approved by the Committee on the Ethics of Animal Experiments of “Pays de la Loire” (permit # CEEA.2011.14). *Entpd1^−/−^* mice were generated as previously described [15] and were backcrossed at least seven times on a C57BL/6J background. Heterozygous mice were crossed and 4 to 8-month-old wild-type (WT, *Entpd1^+/+^*) and *Entpd1^−/−^* male littermates were used in all experiments. 

### 4.2. Evaluation of Vascular Reactivity Using Wire Myography

Animals were sacrificed via CO_2_ inhalation. The thoracic aorta (ThAo), mesenteric arteries (MRA), and left coronary arteries (LCA) were dissected in an ice-cold physiological salt solution (PSS) of the following composition (mmol/L): 130.0, NaCl; 15.0, NaHCO_3_; 3.7, KCl; 1.6, CaCl_2_; 1.2, MgSO_4_; and 11.0, glucose. Pharmacological study was performed on a 2 mm long (aorta, MRA) or ≈1.3 mm (LCA) arterial segments mounted on a wire-myograph (DMT, Aarhus, DK) [61]. Maximal contraction was evaluated using KCl (80 mM) and arterial contraction was expressed as a percentage of the maximal response for each artery (%KCl). The cumulative concentration-dependent contraction was built using phenylephrine (Phe) for ThAo and MRA or serotonin (5HT) for LCA (which does not respond to Phe) as the vasoconstrictor. The cumulative concentration–response curve to acetylcholine (Ach) and ATP was performed on pre-contracted (70–80% of maximal contraction: Phe 10^−6^–3.10^−6^ M; 5HT 10^−6^–10^−5^ M) arteries. The contribution of NO and cyclooxygenase-derived prostanoids on the vascular response was evaluated after incubation with the NO synthase inhibitor N(ω)-nitro-L-arginine methyl ester (LNAME 10^−4^ M, 30 min) alone or in combination with the cyclooxygenase inhibitor indomethacin (INDO 10^−5^ M, 25 min). Endothelium-independent relaxation was evaluated at the end of the protocol in response to the NO donor sodium-nitroprusside (SNP). The data were expressed as a percent relaxation of the precontracted artery.

### 4.3. Evaluation of Flow-Mediated Dilation Using Pressure-Myography 

Flow-mediated dilation (FMD) was evaluated using a pressure myograph as previously described [26]. Second-order mesenteric arteries (internal diameter of 140–220 μm) were cannulated at both ends between two glass pipettes and bathed in 37 °C PSS (pH 7.4, oxygenated with 12% O_2_, 5% CO_2_, and 83% N_2_) in a video-monitored perfusion system (Living System Burlington, VT, USA). Perfusion of arterial segments was set with a peristaltic pump under the control of a pressure servo control system. Diameter changes were recorded continuously. Pressure was set at 60 mmHg. After a 40 min period of equilibration, FMD was measured in response to the stepwise increases (0 to 150 µL per min) in the intraluminal flow on pre-contracted arteries (Phe 30 µmol/L). Maximal diameter was evaluated at the end of the experiment by replacing the arterial bath with PSS without Ca^2+^ in the presence of EGTA (2 mmol/L). The data were expressed as a percent relaxation of the phenylephrine-induced contraction.

### 4.4. In Vivo Flow-Dependent Arterial Remodeling

Both outward and inward arterial remodeling was appreciated in the MRA ligation models as previously described [62,63]. Briefly, adult male mice were anesthetized via isoflurane inhalation (2.5% isoflurane in 0.2 L/min of air) and the body temperature was maintained at 37.5 °C using a thermostatically controlled heating platform. A medial laparotomy was performed and a section of the ileum was extracted and spread over a gauze swab that had been dampened with a sterile physiological salt solution. A segment of a first-order mesenteric artery side branch was gently freed of fat and connective tissue under a dissection microscope. Blood flow was decreased in the first-order low flow (LF) artery via the ligation of the downstream second-order branches with 6-0 silk surgical threads. Blood flow was increased in the high flow (HF) artery after the ligation of the two adjacent arteries. Two arteries distant from the ligation served as internal control/normal flow (NF) arteries (see scheme in Figure 4B). Similar surgeries without ligations were performed in the Sham group. Mice were caged and had free access to food and tap water until their use 15 days later. Arterial diameter was measured ex vivo with a pressure myograph on the isolated and cannulated mesenteric arteries, in the absence of calcium and in the presence of EGTA (2 mmol/L). Viscoelasticity parameters were calculated thanks to the continuous measurement of arterial walls and internal diameters following the step-increase in internal pressure (10 to 125 mmHg).

### 4.5. Ex Vivo Working Heart Perfused in Semi-Recirculating Mode

Hearts were isolated and perfused under normoxic conditions for 30 min to evaluate basal function. All perfusions were carried out at a fixed preload (15 mm Hg) and afterload (50 mm Hg), using a semi-recirculating modified Krebs–Henseleit buffer. Functional parameters were monitored throughout the perfusion (iox2 data acquisition software, EMKA Technologies). Hemodynamic parameters were continuously monitored as the atrial and aortic flow rates (entering and outgoing flows) were monitored with calibrated electromagnetic flow probes. Coronary flow was calculated as the difference of the entering and outgoing flow. The left ventricular functions were monitored such as heart rate (HR), maximum and minimal left ventricular systolic and diastolic pressures (Pmax, Pmin), left ventricular end-diastolic pressure (LVEDP), and maximum and minimum value for the first derivative of the maximum left ventricular systolic pressure (dP/dt max) and minimal left ventricular diastolic pressure (dP/dt min). After recording the baseline parameters, the measurements on the working hearts were further assessed on the paced hearts where the cardiac frequency was set at 500 beats per minute.

### 4.6. In Vivo Telemetry 

Radio-telemetric devices (TA11PAC20, Datasciences, Minneapolis) were implanted into the carotid artery of adult male *Entpd1^+/+^* mice and age-matched *Entpd1^−/−^* mice under isoflurane anesthesia (5% in O_2_, 1 L/min for induction and 2% for maintenance), and were instrumented with OpenHeart single-channel telemetry transmitters (Data Sciences International, Arden Hills, MN, USA). For analgesia, buprenorphine (0.05 mg/kg) was administered before and every 8 h for 48 h after surgery. Mice were provided with food and water ad libitum and maintained on a 12:12 h day–night cycle. One week after surgery, the data acquisition was initiated at a sampling frequency of 1 kHz and maintained for 48 h. Recordings were analyzed with ECG Auto (version 2.8; EMKA Technologies, Paris, France). The analyst was blinded as to the genotype, and the mice shared the same library of typical traces.

### 4.7. Millar Catheterization

Mice were anesthetized with 2% isoflurane. Pressure and left ventricular function were measured with a microtip pressure transducer catheter (1.4 French; Millar Instruments, Houston, TX, USA), which was advanced through the carotid artery to the left ventricle. The data were analyzed with the program IOX version 1.8.0 (EMKA Technologies, Falls Church, VA, USA).

### 4.8. Data Analysis

Multiple groups were compared by using a two-way analysis of variance (ANOVA) with or without repeated measures followed by a post-test Bonferroni’s correction. Two groups were compared by using an unpaired Student’s *t* test (two-tailed); *p* values of <0.05 were considered statistically significant. 

## 5. Conclusions

In addition to its role in the maintenance of the anti-thrombotic and anti-permeability properties of the vascular endothelium, CD39 also contributes to endothelial mechanosensing of shear stress by maintaining proper P2 receptor reactivity. The present paper contributes to feed a growing body of evidence suggesting that purinergic signaling is involved in arterial diameter adaptation in response to mechanical forces, namely the myogenic tone and FMD. Pathological context associated with the alteration in the CD39 expression may affect vascular plasticity via impaired endothelial mechanotransduction. 

## Figures and Tables

**Figure 1 ijms-24-15038-f001:**
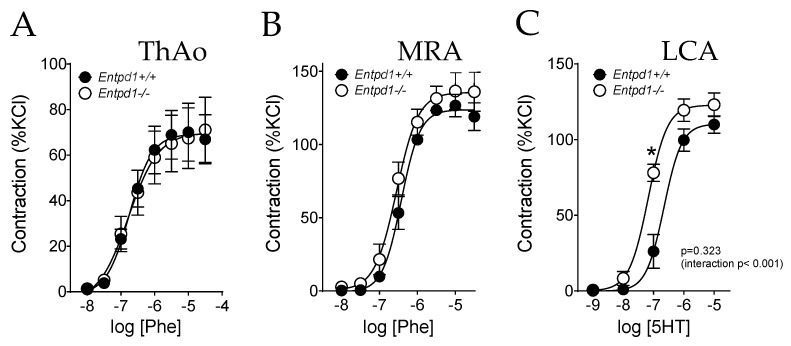
Impact of Entpd1 deficiency on contracting responses of conduit (ThAo) and resistance (MRA and LCA) arteries in response to pharmacological agonists. Aortic ((**A**); ThAo), mesenteric ((**B**); MRA) and coronary ((**C**); LCA) segments were isolated from male *Entpd1^+/+^* and *Entpd1^−/−^* mice (n = 5 mice per group) and mounted on a wire-myograph. Phenylephrine (Phe) (ThAo and MRA) and serotonin (5HT) (LCA) induced concentration-dependent contractile responses. Means ± SEM are shown. Two-way ANOVA for repeated measurements *Entpd1^+/+^* vs. *Entpd1^−/−^*, *p* values are shown on each graph such as Bonferroni post-test * *p* < 0.05 *Entpd1^+/+^* vs. *Entpd1^−/−^*.

**Figure 2 ijms-24-15038-f002:**
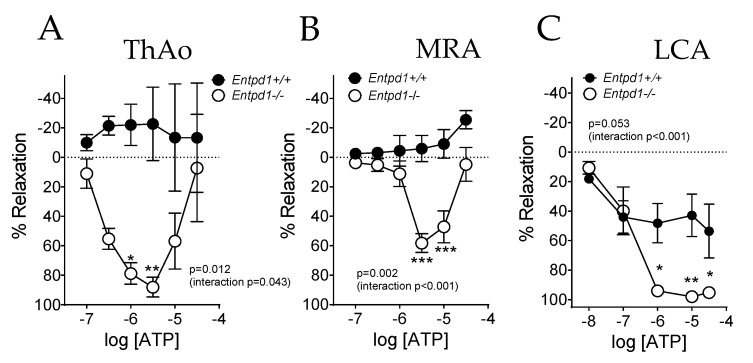
Impact of Entpd1 deficiency on vascular responses of conduit (ThAo) and resistance (MRA and LCA) arteries in response to ATP. Aortic ((**A**); ThAo), mesenteric ((**B**); MRA) and coronary ((**C**); LCA) segments were isolated from male *Entpd1^+/+^* and *Entpd1^−/−^* mice (n = 5 mice per group) and mounted on a wire-myograph. Increasing cumulative concentrations of ATP on pre-contracted arteries induced a bimodal response resulting in a major relaxing response in *Entpd1^−/−^* mice. Means ± SEM are shown. Two-way ANOVA for repeated measurements *Entpd1^+/+^* vs. *Entpd1^−/−^*, *p* values are shown on each graph such as Bonferroni post-test * *p* < 0.05 and ** *p* < 0.01, *** *p* < 0.001 *Entpd1^+/+^* vs. *Entpd1^−/−^*.

**Figure 3 ijms-24-15038-f003:**
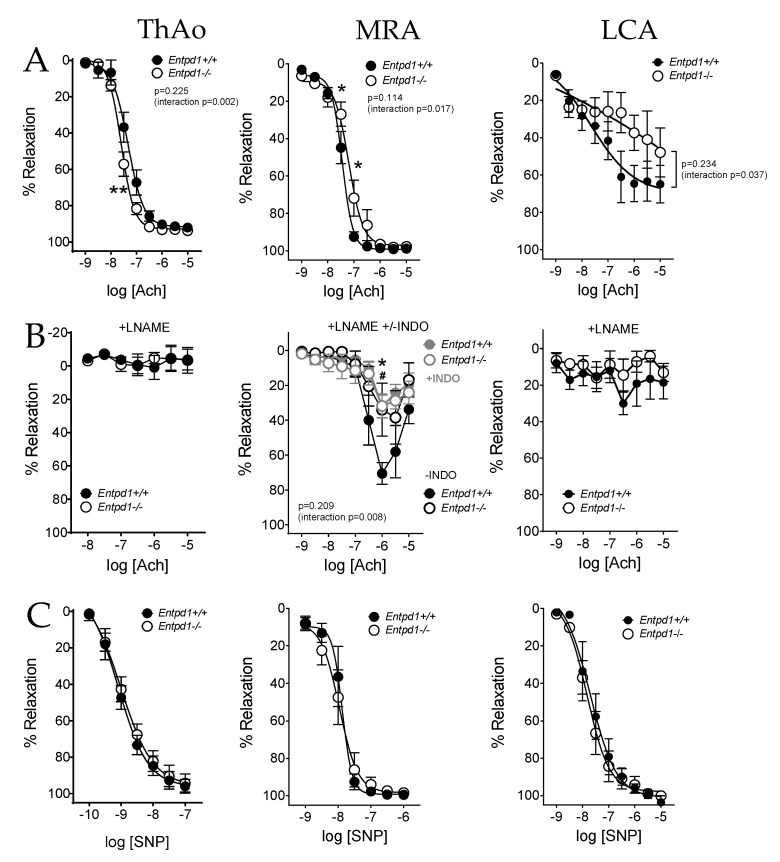
Impact of Entpd1 deficiency on relaxing responses of conduit (ThAo) and resistance (MRA and LCA) arteries. Aortic (ThAo, left), mesenteric (MRA, middle) and coronary (LCA, right) segments were isolated from male *Entpd1^+/+^* and *Entpd1^−/−^* mice (n = 5 mice per group) and mounted on a wire-myograph. (**A**) Endothelium-dependent relaxations were assessed in response to increasing cumulative concentrations of Ach on pre-contracted arteries. (**B**) NO-independent relaxations were measured in the presence of NOS antagonist LNAME (10^−4^ M) alone or with indomethacin (+INDO 10^−5^ M, in gray) for further COX-inhibition. (**C**) Endothelium-independent relaxation was assessed in response to the NO donor SNP. Means ± SEM are shown. Two-way ANOVA for repeated measurements *Entpd1^+/+^* vs. *Entpd1^−/−^*, *p* values are shown on each graph such as Bonferroni post-test * *p* < 0.05 and ** *p* < 0.01 *Entpd1^+/+^* vs. *Entpd1^−/−^*; # *p* < 0.05 *Entpd1^+/+^* LNAME vs. *Entpd1^+/+^* LNAME+INDO.

**Figure 4 ijms-24-15038-f004:**
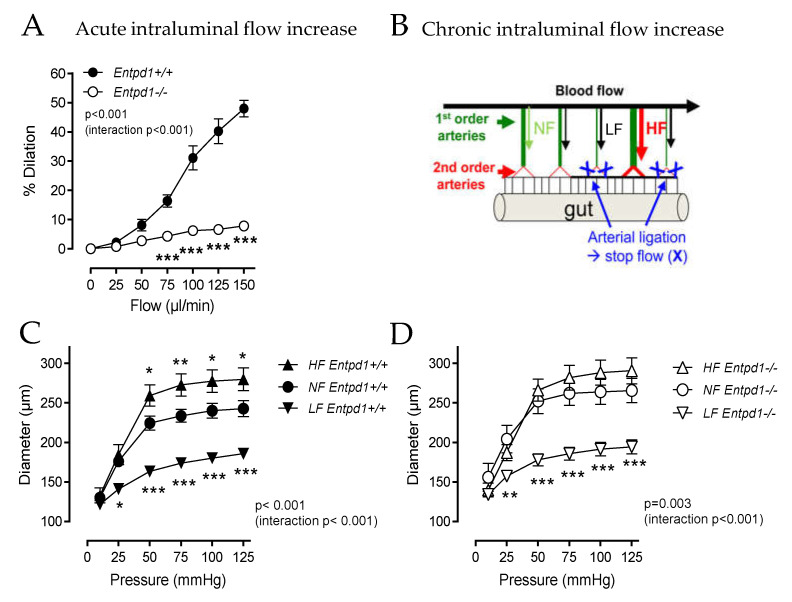
MRA vascular response to acute and chronic increase in intraluminal flow. (**A**) Flow-mediated dilation was measured in mesenteric resistance arteries isolated from male *Entpd1^+/+^* and *Entpd1^−/−^* mice (n = 7 mice per group). (**B**) Schematic representation of the protocol used to induce the flow-mediated remodeling of mesenteric arteries in vivo (see Materials and Methods). Following ligation of some second-order mesenteric arteries downstream the first-order mesenteric arteries, first-order arteries with ligation are submitted chronically to high flow (HF) and compared with the first-order mesenteric arteries without ligation (normal flow, NF) and with ligation (low flow, LF). (**C**,**D**) 2 weeks after surgery, flow-mediated remodeling was evaluated by the measurement of passive diameter in mesenteric arteries isolated from WT *Entpd1^+/+^* (**C**) and *Entpd1^−/−^* (**D**) mice in response to stepwise increases in intraluminal pressure (n = 6 per group). Means ± SEM are shown. two-way ANOVA for repeated measurements ((**A**): *Entpd1^−/−^* versus *Entpd1^+/+^* mice; (**C**,**D**): HF and LF versus NF control; *p* values are shown on each graph) and Bonferroni post-test were performed * *p* < 0.05, ** *p* < 0.01, *** *p* < 0.001.

**Figure 5 ijms-24-15038-f005:**
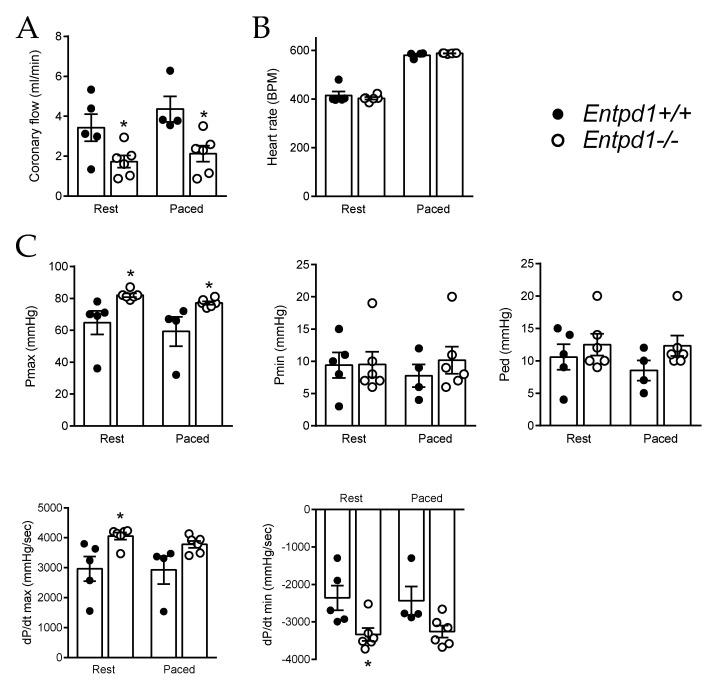
Impact of Entpd1 deletion on isolated perfused heart. (**A**) Coronary output, (**B**) heart rate and (**C**) left ventricular functions were evaluated on isolated pressurized and perfused hearts of *Entpd1^−/−^* (white circles) and *Entpd1^+/+^* (black circles) littermate mice. Systolic function was evaluated through maximal pressure (Pmax) and maximal change in pressure over time (dP/dt max) and diastolic function through minimal (Pmin), end-diastolic pressures (Ped) and minimal change in pressure over time (dP/dt min). Measurements were performed either in resting conditions (Rest) or on paced working heart (Paced). Means ± SEM are shown. Regular two-way ANOVA ((**A**–**C**): *Entpd1^−/−^* versus *Entpd1^+/+^* mice) and Bonferroni post-test were performed. * *p* < 0.05 *Entpd1^−/−^* vs. *Entpd1^+/+^* mice.

**Table 1 ijms-24-15038-t001:** Hemodynamic parameters assessed on conscious mice via radiotelemetry. MAP: mean arterial pressure; SBP: systolic blood pressure; DBP: diastolic blood pressure; PP: pulse pressure; HR: heart rate. a.u.: arbitrary unit. Student *t*-test * *p* < 0.05.

Variable	*Entpd1^+/+^*	*Entpd1^−/−^*	*p* Value
n	8	9	
MAP (mmHg)	109 ± 2	107 ± 1	0.168
SBP (mmHg)	126 ± 2	121 ± 1	* 0.018
DBP (mmHg)	90.0 ± 1.4	91.5 ± 1.4	0.471
PP (mmHg)	36.3 ± 0.9	29.7 ± 2.2	* 0.019
HR (beats/min)	564 ± 10	534 ± 6	* 0.022
Activity (a.u.)	4.77 ± 0.43	5.72 ± 0.87	0.365

**Table 2 ijms-24-15038-t002:** Hemodynamic parameters assessed on anesthetized mice (intra-cardiac and intra-carotidian Millar pressure probe). MBP: mean blood pressure; SBP: systolic blood pressure; DBP: diastolic blood pressure; PP: pulse pressure; Pmax, Pmin and Ped: maximal, minimal, and end-diastolic left ventricular pressures; dP/dt max and dP/dt min: maximal and minimal changes in left ventricular pressure over time. CT: contraction time; RT: relaxation time; Tau: relaxation time constant; HR: heart rate. Student *t*-test * *p* < 0.05.

Variable	*Entpd1^+/+^*		*Entpd1^−/−^*		*p* Value
**n**	**5**		**6**		
MBP (mmHg)	70.0	±2.9	79.3	±4.6	0.144
SBP (mmHg)	88.4	±2.1	94.2	±4.1	0.263
DBP (mmHg)	54.2	±3.9	67.2	±4.5	0.064
PP (mmHg)	34.2	±2.2	27.1	±0.9	* 0.011
Pmax (mmHg)	93.8	±2.3	95.0	±2.6	0.740
Pmin (mmHg)	3.88	±0.71	5.50	±0.83	0.182
Ped (mmHg)	6.98	±0.71	9.17	±1.13	0.154
CT (msec)	11.6	±0.4	11.8	±0.4	0.693
RT (msec)	43.7	±0.6	42.2	±0.5	0.063
dP/dt max (mmHg/s)	6698	±327	6762	±489	0.919
dP/dt min (mmHg/s)	−5557	±126	−5196	±318	0.353
Tau (msec)	7.60	±0.51	6.50	±0.34	0.098
HR (beats/min)	401	±19	460	±15	* 0.035

**Table 3 ijms-24-15038-t003:** Pharmacological vascular response to vasoconstricting (phenylephrine (Phe) and serotonin (5HT)) and vasodilating (acetylcholine (Ach)) agents of aortic (ThAo), mesenteric (MRA), and coronary arteries (LCA). EC_50_ (M), E_max_ (% contraction or % relaxation). Mann–Whitney test * *p* < 0.05.

Variable	*Entpd1^+/+^*	*Entpd1^−/−^*	*p* Value
**n**	**5**	**5**	
**ThAo**			
Phe EC_50_	2.04 × 10^−7^ ± 3.89 × 10^−8^	1.90 × 10^−7^ ± 2.92 × 10^−8^	0.944
Phe E_max_	70.73 ± 11.62	70.2 ± 13.94	0.532
Ach EC_50_	4.98 × 10^−8^ ± 1.11 × 10^−8^	2.70 × 10^−8^ ± 4.07 × 10^−9^	0.222
Ach E_max_	91.41 ± 1.42	92.8 ± 2.12	0.532
**MRA**			
Phe EC_50_	4.04 × 10^−7^ ± 5.53 × 10^−8^	3.24 × 10^−7^ ± 7.61 × 10^−8^	0.532
Phe E_max_	127.06 ± 6.87	138.92 ± 12.89	0.532
Ach EC_50_	3.65 × 10^−8^ ± 6.57 × 10^−9^	1.04 × 10^−7^ ± 5.09 × 10^−8^	0.095
Ach E_max_	99.02 ± 0.57	98.39 ± 0.24	0.532
**LCA**			
5HT EC_50_	2.61 × 10^−7^ ± 7.30 × 10^−8^	7.14 × 10^−8^ ± 5.39 × 10^−8^	* 0.029
5HT E_max_	110.30 ± 5.86	123.65 ± 8.47	0.486
Ach EC_50_	6.75 × 10^−8^ ± 5.13 × 10^−8^	3.10 × 10^−7^ ± 2.13 × 10^−7^	0.873
Ach E_max_	67.84 ± 10.62	54.71 ± 13.89	0.524

## Data Availability

The data presented in this study are available on reasonable request to the corresponding author.

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
