# Peer review of "NTPDase1/CD39 Ectonucleotidase Is Necessary for Normal Arterial Diameter Adaptation to Flow"

_ijms, 2023, doi:10.3390/ijms242015038_

Round 1

Reviewer 1 Report

In their study, the authors examine the importance of CD39, an ectonucleotidase involved in ATP and ADP degradation and thromboinflammation, for vascular function regulation. For this, they use mice with global CD39 deficiency and examine systemic blood pressure levels using wire and pressure myography, telemetry as well as Millar catheterization, and ex vivo working heart and mesentery artery ligation models.

In general, the manuscript is well-written and easy to follow. The data are well presented and the conclusion supported by findings.

 Comments:

1.       The majority of data are vascular function and blood pressure data. The study would benefit from some histological analysis of the vessel types studies, in particular related to factors affecting vascular reactivity, such as P2 receptor and eNOS expression or the levels of oxidative stress.

2.       The differential response of different vascular beds and their dependency of eNOS is interesting. How do the findings relate to differences in CD39 expression in different areas of the vasculature? Do the authors have any insights into this aspect?

3.       Were endothelial cells isolated from the mice and differences in their gene expression profile analysed? If not, the authors could discuss this point. How were endothelial P2 receptor levels altered? Were markers of endothelial dysfunction elevated (e.g. VCAM1, VWF or ET-1)?

4.       What would be the histological correlate of increased arterial compliance? Does the arterial wall differ in size or composition?

5.       Figure 5 (seems to be misplaced and is also not discussed in the text) and regarding the observed changes in coronary blood flow: can the authors discuss possible consequences of this observation or show histological analysis of cardiac vessels, immune cardiac fibrosis.

6.       Also, and regarding the observed changes in cardiac output, the authors should discuss the possibility that cardiac output changes may have affected the vasculature.

7.       The manuscript also would benefit from some histological images of the heart and the cardiac vasculature, and immunohistochemical or flow cytometry analysis of changes in the perivascular space or infiltrating immune cells also would be of interest.

8.       The authors should introduce or discuss aspect of the phenotype of CD39 knockout mice (Endtpd1-/-), in particular related to changes in platelet numbers and reactivity, and how they may have affected the present findings. In this regard, the authors should also discuss physiological or pathological conditions in which ATP concentrations are elevated (and CD39 of relevance).

9.       Although I made some suggestions for the discussion, this sections needs to be shortened and better focussed on the results. It’s quite lengthy.

10.   I also would suggest to modify the title and replace the term “arterial remodeling” to avoid any confusion with vascular remodeling during atherosclerosis or neointima formation. It also does not reflect the data obtained from the acute experiments performed in this study.

Reviewer 2 Report

The work was done by a group highly skilled in measuring cardiovascular functions in vivo and ex vivo.  This is basically a data paper and as such, lacks hypothesis-driven approaches, which are needed to elucidate biological mechanisms.  The KO animal model used in the study is a global KO and as such, various observed effects on the cardiovascular system of the KO mice cannot be attributed to the absence of CD39 in any specific cardiovascular cell type.  In this regard, the discussion is speculative.  It is also very long.  The authors are suggested to reduce the length of discussion by at least 50% by reducing speculative biological discussion.  I see that the strengths of the paper is the technical aspects.

The authors wanted to know the role of CD39 expressed on the cells of the cardiovascular system in cardiovascular hemodynamics.  They used global CD39 KO mice for the study, but this is the wrong model to use. What they need to do is to knockout specifically in the cells of the cardiovascular system and do the study.  This requires several cell-specific inducible mouse models. What the study shows is not what the authors aimed for, but it shows the hemodynamic properties of global CD39 KO mice.  The authors interpret all the data as though the effects are due to the CD39 null condition of the cardiovascular cells, which is not true.  The effects are due to the global KO (not cell specific) of CD39, not to the null condition of any specific cell type.

The study would be very relevant if there is a human disease due to the global CD39 null condition.  I do not know the answer to this, but this condition may be rare.  The use of cell-specific, inducible animal models is required to get meaningful/significant data.  Another issue is that the authors argue that CD39 is a critically important molecule for the proper functioning of the cardiovascular system and deserves close attention.  This argument seems to be an exaggeration as mice with no expression of this protein can live well, showing the molecule is not required for life.  The results from the global CD39 KO mouse indicates that CD39’s role in the function of the cardiovascular system is at best accessory, not essential.

The study adds new data on the hemodynamics of systemic CD39 null mice, but it is unclear how useful such data are.  In general, data from global KO animals are appreciated less and less.  The work would have been wonderful 15 years ago, but in 2023, the authors should have used cell-specific inducible models.

Round 2

Reviewer 1 Report

Thank you for addressing all of my comments, I have no further suggestions. 

Author Response

We thank the reviewer for his questions/comments that helped us to improve our manuscript.

Reviewer 2 Report

The author changed the title, which is in line with the way the study was done.

In the rebuttal, the authors seem to suggest that smooth muscle CD39 plays no role in all the responses they reported.  This is an overstatement as there are many papers that report roles of CD39 in smooth muscle contraction and cardiovascular diseases. For example:

“NTPDase1 modulates smooth muscle contraction in mice bladder by regulating nucleotide receptor activation distinctly in male and female”. Romuald Brice Babou Kammoe, Gilles Kauffenstein, Julie Pelletier, Bernard Robaye and Jean Sévigny. Biomolecules 2021, 11(2):147. doi: 10.3390/biom11020147

NTPDase1 (CD39) controls nucleotide-dependent vasoconstriction in mouse. Gilles Kauffenstein, Annick Drouin, Nathalie Thorin-Trescases, Hélène Bachelard, Bernard Robaye, Pedro D'Orléans-Juste, François Marceau, Eric Thorin, Jean Sévigny. Cardiovasc Res. 2010. 85(1):204-13. doi: 10.1093/cvr/cvp265

These papers are authored by some of the authors of this submitted paper.  For interpreting the data, the authors must not ignore the possible (or more accurately, probable) contribution of smooth muscle CD39.

On a separate note, I cannot see why Fig. 5 appears before any other figures.  In addition, I believe Fig. 5B is cited in error.

Minor editing.

Author Response

We would like to thank one more time the reviewer for his constructive comments.

We indeed reported a contribution of CD39 expressed in vascular (Kauffenstein et al. 2010) and non-vascular smooth muscle (Kauffenstein et al. 2014; Babou Kammoe et al. 2021). Concerning CD39 expression in VSMC, we found that the enzyme limits ex vivo vasoconstriction in response to nucleotides with potential impact on myogenic and neurogenic inputs in vivo.

The vascular responses we investigated here clearly depend on endothelial responses. These are abrogated after physical disruption or pharmacological inhibition of endothelial function. Moreover, despite a physiological antagonism of myogenic tone and flow dependent vasodilation we reported in different murine models a functional dissociation of the 2 processes (Loufrani et al. 2001; Henrion et al. 1997). Hence, we do think that the phenotype and the defect we report here for Entpd1-/- mice mostly altered endothelial response.

However, we agree with the reviewer that we cannot fully exclude a potential role of VSMC-expressed CD39 and an impact of its deletion in thwarting endothelial-dependent relaxation, especially in long term processes as MRA outward remodeling (ligation model) and decreased coronary blood flow. This hypothesis has been included in the discussion.

In conclusion, we evidence a new role of endothelial CD39 in the regulation of acute response to flow with the possible (although unlikely) contribution of the muscular enzyme in te phenotype of Entpd1-/- animals in chronic setup (isolated heart, high chronic flow).

As already mentioned by the reviewer (now discussed), tissue-specific deletion of the enzyme may help deciphering the mechanisms by which CD39 influences the overall microvascular mechanotransduction.

Concerning the figure 5, we notified its misplacement.

Finally, the mention of Figure 5B has been changed for Figure 4B.

References

Babou Kammoe, Romuald Brice, Gilles Kauffenstein, Julie Pelletier, Bernard Robaye, et Jean Sévigny. 2021. « NTPDase1 Modulates Smooth Muscle Contraction in Mice Bladder by Regulating Nucleotide Receptor Activation Distinctly in Male and Female ». Biomolecules 11 (2): 147. https://doi.org/10.3390/biom11020147.

Henrion, D., F. Terzi, K. Matrougui, M. Duriez, C. M. Boulanger, E. Colucci-Guyon, C. Babinet, et al. 1997. « Impaired flow-induced dilation in mesenteric resistance arteries from mice lacking vimentin ». J Clin Invest 100 (11): 2909‑14.

Kauffenstein, G., A. Drouin, N. Thorin-Trescases, H. Bachelard, B. Robaye, P. D’Orleans-Juste, F. Marceau, E. Thorin, et J. Sevigny. 2010. « NTPDase1 (CD39) controls nucleotide-dependent vasoconstriction in mouse ». Cardiovasc Res 85 (1): 204‑13.

Kauffenstein, G., J. Pelletier, E. G. Lavoie, F. Kukulski, M. Martin-Satue, S. S. Dufresne, J. Frenette, et al. 2014. « Nucleoside triphosphate diphosphohydrolase-1 ectonucleotidase is required for normal vas deferens contraction and male fertility through maintaining P2X1 receptor function ». J Biol Chem 289 (41): 28629‑39. https://doi.org/10.1074/jbc.M114.604082.

Loufrani, L., K. Matrougui, D. Gorny, M. Duriez, I. Blanc, B. I. Levy, et D. Henrion. 2001. « Flow (shear stress)-induced endothelium-dependent dilation is altered in mice lacking the gene encoding for dystrophin ». Circulation 103 (6): 864‑70.